# Community Pharmacies in the Asian Countries of Developing Health System: Formation, Regulation, and Implication

**DOI:** 10.3390/pharmacy11040127

**Published:** 2023-08-10

**Authors:** Shaymaa Alenezi, Mona Alanazi, Reaam Aljazaeri, Marwah Almuzaini, Shorouq Alrasheidi, Wafa Bin Shamlan, Raghad Aljohani, Ghadeer Alhawiti, Meaad Alqarni, Ehdaa Aljabri, Majd Qmmash, Mohammed Kanan

**Affiliations:** 1Al-Dawaa Pharmacy, Raiydh 12211, Saudi Arabia; ph.shyma33@gmail.com (S.A.); marwahalmuzaini.410@gmail.com (M.A.); shorouqmohammad23@hotmail.com (S.A.); 2Department of Clinical Pharmacy, Nahdi Medical Co., Riyadh 13515, Saudi Arabia; ph.monah@gmail.com (M.A.); reaamaljazaeri@gmail.com (R.A.); raghadjohani@outlook.com (R.A.); meaadgarnii@gmail.com (M.A.); ehdaa.aljabri@gmail.com (E.A.); 3Community Pharmacist, United Pharmaceutical Company, Jeddah 22230, Saudi Arabia; wafabinshamlan@gmail.com; 4Department of Clinical Pharmacy, Alkadi Medical Group, Tabuk 47311, Saudi Arabia; ghadeerab98@gmail.com; 5College of Pharmacy, King Khalid University, Abha 62529, Saudi Arabia; majdqh1464@gmail.com; 6Department of Clinical Pharmacy, King Fahad Medical City, Riyadh 12211, Saudi Arabia

**Keywords:** developing countries, Asia, community pharmacies, formation

## Abstract

Community pharmacies (CPs) in Asian countries are often the first point of contact for patients withinthe healthcare system and their preferred place to purchase medicines. The number of CPs may vary across Asian countries, and each country has developed its own design and functioning. The regulatory environment plays a crucial role in shaping and governing CPs. The aim of this study was to conduct a comprehensive literature review in order to enhance understanding of the establishment and regulation of CPs. By undertaking this review, the researchers, policymakers, and practitioners sought to gain a deeper insight into the performance and functions of CPs and the regulatory frameworks that govern them. Moreover, this review shed light on implementation strategies, effects on patient outcomes, and the barriers and challenges associated with their establishment. A narrative literature review method was adopted with specific inclusion and exclusion criteria. Significant disparities can be observed when comparing the stated intentions of regulations with their actual implementation. Recently, there has been an inclusion of public health practices. Unfortunately, pharmacy procedures conducted in such environments have been characterized by inadequate understanding and inappropriate care. This poor performance can be attributed to employees’ focus on maximizing profits. Several shortcomings can arise, including incomplete patient history documentation, failure to refer patients who require medical attention, unauthorized dispensing of prescription-only medicines (POM), dispensing clinically inappropriate or excessive medication doses, selling incomplete antibiotic courses, and inadequate information and counseling services. Regulatory interventions can help strengthen these services.

## 1. Introduction

Community pharmacies (CPs) are widely recognized as highly accessible healthcare facilities that offer convenient access to primary care services and a wide range of healthcare products [1]. CPs offer a diverse range of pharmaceutical services with the aim of enhancing individuals’ overall health and well-being. These services are designed to provide personalized care and attention to patients and are typically delivered by trained pharmacists [2,3]. A predominant proportion of CPs are under private ownership [4,5,6], and the financial stability of the concerned entity remains dependent on the delivery and distribution of medicinal substances. Currently, the primary area of emphasis is in their professional practice and commercial operations involvement inthis aspect. The ubiquity of CPs in proximity to residential, occupational, and commercial locations on a global scale has been well established [5,6]. In several developed nations, the trend in CPs services is shifting towards a more patient-centered approach. This is evidenced by the provision of specialized services such as medication therapy management, management of minor ailments, smoking cessation support, vaccination, and medication adherence [7,8]. PC services play a vital role in the delivery of healthcare in developed nations and are often integrated into their healthcare systems [9]. The progression of these developments has been greatly impacted by government policies. The primary goals of this initiative are to promote optimal healthcare service utilization and to optimize the responsible and judicious use of medications, leading to improved safety, effectiveness, and efficiency [10]. According to the survey conducted by the International Pharmaceutical Federation (FIP) in the period of 2020–2021, there are approximately 1.6 million CPs across 76 countries globally, catering to approximately 75% of the global population. Variations in the scope and categories of CPs are observable across nations, with extensive chain establishments in metropolitan areas and modest autonomous stores in rural or suburban localities [11]. In developing countries, it continues to be in the product-oriented practice stage [12]. In most developing countries, CPs are primarily engaged in drug selling, and the establishment of PC is not fully realized. The laws and policies regulating the supply of medicines from CPs vary in each country [13,14,15,16,17,18,19,20]. To date, no comprehensive study has been formulated to represent CPs in Asian countries. Therefore, this study was conducted to provide a snapshot of CPs. The objective of this narrative review was to present a comprehensive review of the relevant literature and identify key studies that demonstrate the features of pharmacy services, organizational structure, and legislative frameworks in community settings.

## 2. Methods

A comprehensive review of the existing literature was conducted through an extensive search using prominent databases such as Cochrane, PubMed (Medline), Google Scholar, Scopus, Science Direct, and Springer Links. A narrative literature review encompasses a comprehensive exploration, evaluation, and synthesis of scholarly articles and textbooks centered around a specific subject.

The present study also included a thorough search of grey literature to identify any unpublished reports related to CPs reported by nations, organizations, or pharmacy professional associations. Grey literature refers to information created and disseminated through non-conventional publishing and distribution methods. This category includes a wide range of materials such as reports, policy documents, working papers, newsletters, government publications, speeches, white papers, urban plans, and other similar resources. The records were extracted from November 2022 to May 2023. The search terms used for each selected country in Asia included community pharmacies, retail pharmacies, pharmacy shops, compliance with legislation, regulations, pharmacy services, community pharmacists, pharmacy staff, medicine storage, performance of CPs, number of CPs, challenges, limitations in work, low-income countries, developing countries, traditional services (compounding and dispensing), medication management therapy, patients’ care, disease management, public health service, preventive role, and community health. Only articles in English were considered, and potential articles were independently reviewed by each author for inclusion based on the inclusion and exclusion criteria.

### 2.1. Inclusion Criteria

Published in the English language;Randomized controlled trials RCTs; both experimental and non-experimental studies; non-rcts, non-randomized (quasi-experimental) studies; observational, retrospective, and prospective cohort studies; and analytical and cross-sectional studies;Published in peer-reviewed journals, books, and professional websites of authority.

### 2.2. Exclusion Criteria

Low levels of evidence, such as case reports, study protocols, commentaries, or blogs and speeches;Old laws abolished by new ones (if any).

The studies identified through the literature search were screened by threegroups of independent reviewers to assess their eligibility. Group A (SA, MA) was responsible for record gathering for each country. Group B (RA, GA) was responsible for identifyingthe literature about current performance. Group C (RA, EA) was responsible for study screening aboutdeterminants.Any disagreements were resolved through consensus or by a third member fromthe group of reviewers (MQ, WBS).

## 3. Results

### 3.1. Formation of CPs

#### 3.1.1. China

According to the China Food and Drug Administration (CFDA), a total of 472,000 medicine outlets with various capacities and operational natures were licensed [21]. The trend of chain pharmacies became more common after 2009 [22]. The Drug Administration Law, version 2001, has the major jurisdiction over the working of CPs.In 2006, the CFDA declared that ownership and possession of any retail pharmacy is not only restrictedto qualifiedpharmacists but also that the sale of medicine should be under the supervision of a licensed pharmacist [23,24]. PC has some limitations due to professional skills and knowledge. Additionally, only 9% of pharmacy graduates are currently engaged in community pharmacy settings [23]. In rural areas, the practices are not much better. The availability of prescription-only medicines (PoM),such as antibiotics over the counter (OTC),is common [25]. In 2014, the Non-Prescription Medicines Association of China recommended Good Pharmacy Practice (GPP) standards for community pharmacies. However, after five years, GPP certification criteria were met by only 86 pharmacies [26].

#### 3.1.2. India

There are over 600,000 licensed retail outlets for the sale and supply of medicines in India. Chain pharmacies are not much different from individual CPs. Despite this, CPs in India are considered essential and cost-effective healthcare resources. Most of these pharmacies are managed by individuals holding a 2-year diploma in pharmacy [27]. The presence of qualified personnel in CPs is limited to only 50%. Most CPs are owned by unqualified and non-professional individuals. Community pharmacists are often hired at very low salaries. Moreover, a majority of community pharmacists are primarily involved in drug dispensing and lack adequate counseling abilities [28]. The majority of CPs sell the PoM without a prescription.The Pharmacy Act of 1948 and the Drugs and Cosmetics Act and Rules of 1945 serve as the foundation for all legislations that regulate the practices of working pharmacists [29].

#### 3.1.3. Indonesia

After 1975, the paradigm of the pharmacy profession in Indonesia also shifted from productdevelopment to patient care [30]. There are 135,000 CPs in Indonesia, with fivemajor chain pharmacies. The ownership of CPs is unconstrained for pharmacists. The CPs’ services also offer limited pharmacy care. Community pharmacists are engaged in activities such as screening, compounding, packaging, labeling, and counseling. They provide medication and health information, including guidance on healthy nutrition, physical activity, smoking cessation programs, and alcohol consumption [31]. They also play a role in monitoring the use of drugs. However, a lack of public confidence in these services still poses a barrier. The regulation of CPs and employment services falls under the purview of government authorities, as stated in Section 51 of the Act 2009 [30,31].

#### 3.1.4. Iraq

More than 9000 CPs that are under the supervision of licensed pharmacists are widely distributed in Iraq. All CPs are privately owned, and there are no international chain pharmacies present. Pharmacy Act 1923 regulates and controls the sale of medicines and poisons. The Syndicate of Iraqi Pharmacists regulates the practices of these pharmacies [32]. According to the rules, two adjacent CPs in Iraq should not have a distance of less than 25 square meters between them. It is also compulsory for a pharmacist to be present during working hours at the pharmacy [33]. The minimum area requirement for a CP is 20 square meters. The working hours per day may vary from CP to CP, depending on the type of practicing license awarded to the pharmacist. Pharmacists working in public institutes or organizations can obtain a part-time license, allowing the CPs under their supervision to operate after 4:00 P.M. [34]. Along with their traditional roles, Iraqi community pharmacists also provide advanced services such as Medication Therapy Management (MTM) is a term used to describe a broad range of health care services provided by pharmacists. It is a patient-centered process aimed at creating treatment plans that revolve around each patient’s specific medication-related goals. The approach involves pharmacists working closely with patients to optimize their medication therapy, ensure proper medication use, address concerns or issues, and ultimately improve health outcomes. The focus is on tailoring the treatment plan to the individual needs and preferences of each patient [32].

#### 3.1.5. Jordan

More than 2500 pharmacies have beenestablished in Jordan [35]. There are about 11 chain pharmacy systems operating in Jordan. The first chain pharmacy was established in 2001. The largest group among them has 76 branches [36,37]. Chain pharmacies offer a high standard of healthcare through well-trained and qualified pharmacists. These chain pharmacies also provide 24/7 online drug information services, ensuring accessibility to drug-related information at all times. [36]. The Drug and Pharmacy Law of 2001 is the foundation of jurisdictions that regulate CPs. Dispensing medications and providing medication counseling, especially for OTC medicines, are integral parts of the work of community pharmacists [38].Some barriers exist in the implementation of the “Medication Review” process [14].

#### 3.1.6. Malaysia

In Malaysia, there are approximately3300 pharmacies [39]. The Poison Act of 1952 has jurisdiction over the functioning of CPs in Malaysia. The MoH has defined requirements for CPs in terms of their area, infrastructure, personnel, practice, and equipment [40]. The presence of a Type“A” licensed pharmacistis necessary for drug dispensing in CPs. The minimum area requirement for a CPis 200 square feet, with specific areas designated for waiting, dispensing, and counseling purposes. CPs operate as independent pharmacies, chain pharmacies, or attached to clinics. Barriers to GPP includetime shortage, lack of human resources and manpower, a collaboration deficit between healthcare personnel, and self-inefficiency, along withregulatory and legal constraints [39,40].

#### 3.1.7. Nepal

There are 14,899 CPs in Nepal [41]. It is common for non-pharmacists to run most CPs. Pharmacists are typically allowed to own a single pharmacy. The operating hours of a CP may vary based on the number of patients visiting the pharmacy [42]. In addition to medications, some CPs also sell cosmetic and general items. By law, the sale of PoM is restricted to a doctor’s prescription. However, it is true that many pharmacies violate this law by selling PoM without a prescription [43,44]. All medicine outlets, i.e., chain and individual, work similarly [44]. Only a few CPs provide MTM services. The Department of Drug Administration serves as the controlling body for CPs’ practice, operating under the Drug Act of 1978 and the national drug policy of 1995 [42,45].

#### 3.1.8. Pakistan

Approximately 80% of medicines are sold through 63,000 CPs that are under the supervision of 810 pharmacists [46]. GPP can indeed be a challenge. It involves ensuring the proper handling, storage, dispensing, and counseling of medications, as well as maintaining high standards of professionalism and patient care. Implementation of GPP includes adequate training, infrastructure, regulatory frameworks, and ongoing monitoring to overcome the challenges and ensure the provision of safe and effective pharmaceutical services [47]. Medication counseling services are often found to be insufficient. This can be attributed to various factors, including a lack of trained pharmacists, limited time available for counseling due to high patient volumes, and inadequate emphasis on patient education within the healthcare system. Efforts are being made to address this issue by promoting the role of pharmacists in medication counseling and enhancing their training and capacity to provide comprehensive Pharmaceutical Care (PC) services. PC encompasses the patient-centered MTM for optimal outcomes [48]. This insufficiencycould be due to resource constraints, inadequate infrastructure, or a lack of regulatory oversight. Efforts are needed to improve the infrastructure and ensure that CPs have the necessary amenities and apparatus to meet the legal requirements and provide quality pharmaceutical services [49]. Another challenge in CPs is the lack of proper training for staff members [50,51]. Legislation to regulate CPs’ operation includes the “Drug Act 1976” and “Pharmacy Act 1967” [50].Non-pharmacists are allowed to own a pharmacy [51].More than half of all medicines sold in Pakistan are supplied without a prescription, which is concerning for patient safety [52,53]. The chain pharmacy system was established in 1995. All the chain pharmacies operate in urban or suburban areas [54,55].

#### 3.1.9. Palestine

There are approximately 1000 pharmacies that operate in a traditional manner. Unlike some other countries, there is no established chain of CPs. Each pharmacy operates independently [55]. According to the law, CP should have a designated compounding area. However, the practice of compounding in CPs is awfully limited. This could be due to various reasons, such as a lack of trained personnel, limited demand for compounded medications, or challenges accessing necessary compounding ingredients and equipment [56,57]. CPs are established under the Palestinian pharmacy practice law [57]. It is unfortunately common for PoM, especially antibiotics, to be sold without a prescription [58,59]. CPs are owned by qualified pharmacists, and it is notable that a significant number of these pharmacists are female [55].

#### 3.1.10. Qatar

About 200 pharmacies are working in the community under the supervision of licensed pharmacy technicians and pharmacists [60]. All of the CPs, whether chain or individual, are privately owned. However, it is important to note that these CPs primarily focus on traditional practices and do not offer specialized or advanced services [61,62]. Several concerns have been noted, like insufficient pharmacists, limited time for patient contact, and unsatisfactory knowledge of pharmacists [62].Almeezan Law No. 3 (1983) regulates the working of pharmacists in the community [61].

#### 3.1.11. Saudi Arabia

In 2020, the Kingdom of Saudi Arabia (KSA) had 17,815 community pharmacists, with only 2271 being Saudi nationals. Saudi pharmacists tend to avoid community settings due to lower pay and job satisfaction [63]. Pharmacists dispense and provide counseling services in the community setting. There are two types of medications: OTC and POM. Big chains of CPs exist [64]. Pharmacists are not legally allowed to diagnose or dispense medicines without a prescription. However, the practice of dispensing medications without prescriptions does occur, despite being illegal [65]. Pharmacists tend to strictly follow the law when it comes to controlled/narcotic substances due to the serious legal repercussions associated with their handling [66]. Pharmacists in community settings often originate from other countries in the Middle East or the Far East [67]. Pharmacy ownership laws have undergone frequent changes [64]. MOH enacted a law in 1978 to regulate the CPs. Only Saudi pharmacists are allowed to manage CPs. Non-pharmacists are permitted to co-own a CP, but the management of the pharmacy is restricted to licensed pharmacists [68]. Only 112 (0.38%) community pharmacists are female [63].

Saudi pharmacies sell a wide range of products, including PoM, OTC products, and non-pharmaceutical items such as herbal products (including herbal teas), cosmetics, baby foods, mother and baby care products, supplements, and sports foods [63,68]. Consumers frequently visit local pharmacies for advice on various aspects, including diseases, OTC medications, cosmetics, and prescription refills. Pharmacists play a vital role in providing guidance and assistance to customers in these areas [67]. Due to the ongoing transformation in the healthcare system, the role of CPs has expanded to include clinical services. These services now encompass tasks such as administering vaccinations, providing patient training on medical equipment and devices, and offering medication management support. Community pharmacists are assuming a more proactive role in delivering comprehensive healthcare services to patients. They are not just focused on dispensing medications but also providea wide range of services such as MTM, health education, preventive care, immunizations, and patient counseling. By actively engaging with patients, community pharmacists contribute significantly to the overall healthcare system by offering accessible and holistic care [69]. Typically, CPs are operated by either one or two pharmacists, sometimes with the assistance of a pharmacy assistant. Employees in these pharmacies are typically expected to work for a duration of 8–12 h, six days a week [70].

#### 3.1.12. Sri Lanka

There are a total of 5029 chain CPs, which is significantly higher compared to approximately 2000 individual CPs. The dispensing of medicine is strictly limited to registered pharmacists. According to regulations, only pharmacists who hold a valid license are authorized to dispense medications to patients [71]. It is concerning that only half of the community pharmacies have the presence of a pharmacist. The presence of a qualified pharmacist is crucial to ensure the safe and appropriate dispensing of both PoMand OTC medications, but there is indeed no significant difference between the dispensing of PoM and OTC medications [72]. The regulation and control of medicine sellingand the practice of CPs are primarily overseen by the National Medicines Regulatory Authority (NMRA-2015) and the MoH (Drugs Act No. 27 of 1980). The NMRA is responsible for the registration, licensing, and quality control of medicines, while the MoH establishes and enforces regulations related to CP practice [71]. Overall, CP services can vary in quality and consistency, and in some cases, they may be considered poor and fragmented [72].

#### 3.1.13. Thailand

More than 11,603 community pharmaciesare located in Thailand. About one-third of them are located in the capital area of Bangkok. The Thai Drug Act of 1967 regulates the business and working of CPs [73]. The Community Pharmacy Association plays a vital role in promoting professional education and providing support to community pharmacists throughout the country. The association is dedicated to enhancing the knowledge and skills of pharmacists, enabling them to deliver high-quality healthcare services to the community.

Through various initiatives and programs, the association strives to continuously improve the professional standards and practices of community pharmacists. It organizes educational seminars, workshops, and training sessions to keep pharmacists updated with the latest developments in pharmacy practice, patient care, and medication management.

Moreover, the association serves as a platform for networking and collaboration among community pharmacists, allowing them to exchange ideas, share experiences, and learn from one another. It also advocates for the recognition and value of community pharmacy services within the healthcare system, highlighting the important role that pharmacists play in promoting patient health and medication safety.

CPs face limitations in providing comprehensive dispensing services due to the authority given to physicians for partial dispensing. The sale of narcotics and certain other controlled drugs requires strict adherence to regulations and record-keeping for the FDA [74]. Some CPs are involved in the screening of chronic diseases such as cardiovasculardiseases, diabetes, hypertension, and osteoporosis. These CPs offer additional services beyond traditional dispensing, focusing on preventive care and disease management [75,76,77,78,79]. Accredited CPs have taken the initiative to provide additional services such as medication therapy management, smoking cessation programs, and prescription refilling. These services aim to enhance patient care, promote medication safety, and support individuals in their efforts to quit smoking. There are regulations in place that restrict the sale of antibiotics to Class 1 medical stores while prohibiting lower-class medical stores from selling them. However, it appears that the enforcement and implementation of this practice may not be consistently followed. A limited number of chain pharmaciesserve across Thailand [80,81,82].

#### 3.1.14. Yemen

A total of 3315 community pharmacies are operational. Very limited domestic chain pharmacies are growing. Law # 26 of 2002 on Medical and Pharmaceutical Professions has major jurisdiction [83]. The typical working hours for CPS range from 8 to 12 h per day. Most CPs observe Friday as a common holiday, aligning with cultural and religious practices. By law, it is important to ensure that CPs are operated by qualified pharmacists to ensure the safe and effective delivery of PC; however, it is concerning that non-pharmacists may be operating some of them [84]. The sale of PoM without a prescription is a concerning practice that can pose significant risks to public health. Antibiotics, analgesics, antipsychotics, and cardiovascular medications are potent medications that are being sold without a prescription. The engagement of non-qualified and unauthorized staff in the diagnosis of illnesses, prescribing medications, and dispensing of medicines for various diseases is significantly common [83,85].

The formation and characteristics of CPs are given in Table 1.

A comparison of density among the counties is given in Figure 1

### 3.2. Challenges Associated with CPs’Services

In the healthcare systems of low-middle-income countries (LMICs), CPs serve as the first point of contact for patients.They are sovereign states characterized by a less developed industrial base and a lower Human Development Index compared to other nations. In these countries, patients frequently show a preference for acquiring medicines through these alternative channels. Unfortunately, the performance of pharmacies in LMICs is characterized by a lack of staff knowledge and inadequate treatment outcomes. Over the past three decades, poor performance has persisted in CPs of LMICs [86].

#### 3.2.1. Prescriptions Filling

Several instances of poor performance have been reported in which prescriptions are not adequately authenticated by drug sellers [87]. It is worth noting that the acceptance of previous prescriptions, even those that are several years old, raises concerns regarding patient safety and the appropriateness of medication use. Prescriptions, especially for certain medications, may have specific validity periods based on medical guidelines and regulations [88,89]. Returning dispensed prescriptions to clients for prospective reuse is a practice that raises concerns regarding patient safety and appropriate medication use. Prescriptions are typically meant to be used for a specific course of treatment, and reusing them without proper medical evaluation can lead to potential risks.

Prescriptions are legal documents that authorize the dispensing of specific medications for a particular individual and condition. Once a prescription has been fulfilled and the medication dispensed, it is generally not advisable to reuse the same prescription for future medication needs [90]. It has been observed that improper dispensing practices are prevalent in several Southeast and South Asian countries. These practices can include dispensing medications without appropriate prescriptions, dispensing prescription-only medicines without proper authorization, inadequate labeling and packaging of medications, and lack of adherence to quality control standards. At times, counseling is not prioritized during medicine dispensing in certain regions such as Central Asia, West Asia, North Asia, and some Middle Eastern countries. Financial constraints often prevent patients from purchasing all the prescribed medications, leading to deviations from the anticipated prescription [91,92]. Drug sellers generally do not engage in any interaction with physicians to inquire about or discuss prescribed medicines [86].

#### 3.2.2. Taking Patients’ History

The taking of patients’ medical historiesin pharmacies is lacking. There is a significant limitation in the questioning process to inquire about other associated signs and symptoms [87,88,89,90,91,92]. There is a rare occurrence of pharmacists asking about co-morbidities, allergies, and other therapies [88,89,90,91,92,93,94]. At the extreme end of poor practice, some pharmacists neglect to ask any questions [95,96,97]. A study describes the rapid exchange of tablets for money [89].

#### 3.2.3. Refer for Doctor Checkup

Some pathological conditions such as asthma, cystitis, prolonged diarrhea, sexually transmitted infections, and tuberculosis (TB)require diagnostic testing and specific PoMs. The treatment of these conditions is beyond the capability of a pharmacist [86]. The quality of recommendations provided by community pharmacists in LMICs is generally substandard, with reported rates ranging from 7% to 37% [98,99]. However, in Vietnam, pharmacies have played a significant role in recommending treatment for TB patients, with 46% of patients being recommended treatment by pharmacies. [100].

#### 3.2.4. Drug Selling

The everyday practice of pharmacies in LMICs raises many concerns related to the legal, clinical, and physical aspects of selling medications. The non-prescription sale of PoMs is a common phenomenon in many settings [86]. A wide range of PoMs like antimalarials, antihistamines, antidiarrheals, antibiotics, anti-hypertensives, anti-epileptics, anti-tuberculosis drugs, hypoglycemics, psycho-tropics, sedatives, steroids, and tranquilizers, are freely available without proper control. As a result, a significant portion of these medicines are used irrationally or without appropriate oversight [87,101,102,103,104,105,106,107,108,109,110,111,112].

In Indonesia, it is common to find antibiotics freely available without a prescription at unregistered roadside kiosks [113]. In addition, the sale of medicines from CPs ofLMICs often demonstratespoor practice when it comes to ensuring their clinical appropriateness for specific diseases [109].

On the other hand, the physical condition and appearance of drugs dispensed to patients is a major concern. In many cases, important medicines are provided in loose strips rather than their original packaging, and labeling is rarely done. Furthermore, there is a tendency to mix up various drugs in the same pack during the dispensing process. These practices can lead to confusion, medication errors, and potential harm to patients [112,114,115,116,117]. These practices result in a loss of information and clarity regarding the active ingredients, dosage, and expiry date of the medications. Patients may not have access to important details about their prescribed drugs, making it difficult for them to understand and use the medications correctly. This lack of information increases the risk of medication errors, adverse effects, and potential harm to patients [86]. Furthermore, there are instances where medicines are improperly stored in CPs. This can include inadequate temperature control, improper storage conditions, and a lack of proper inventory management. Improper storage of medications can lead to degradation of the drug’s effectiveness, loss of potency, and potential safety risks to patients [118].

Treatment duration and proper dosage of medications are often not given adequate consideration by community pharmacists in certain Asian countries. The dispensed doses of medicine may not align with the recommended therapeutic range for specific diseases. This can result in medications being dispensed with suboptimal therapeutic doses or even in cases of overdosing. These practices can have negative consequences on patient outcomes and may lead to ineffective treatment or potential harm [119,120,121]. The issue of dispensing sub-therapeutic doses of antibiotics is prevalent in several countries, including Bangladesh, India, the Philippines, Sri Lanka, Thailand, and Vietnam. This practice can contribute to the development of antibiotic resistance and compromise the effectiveness of treatment [122,123,124]. In LMICs, approximately 50% of antibiotics are sold for durations of less than 2 days [125].

#### 3.2.5. Provision of Medication Advice

The staff of CPs in LMICs often provide limited medication advice and counseling to drug purchasers. Studies have shown that only a small percentage, ranging from 2.5% to 70%, of medicine purchasers receive advice on the proper use of medicines. Patients often receive limited information regarding the appropriate dosage, potential drug or food interactions, and possible sideeffects of medications when interacting with pharmacy staff in LMICs [86]. Counseling services related to sexually transmitted diseases (STDs) and contraceptive medicines are often limited in LMICs. Patients seeking information and guidance regarding STD prevention, treatment, and the use of contraceptive methods may not receive the comprehensive counseling they need [126]. However, dietary instructions are provided to only a minority of patients in LMICs. Pharmacies often do not prioritize or provide sufficient guidance on dietary considerations when dispensing medications [127]. Thus, the staff of CPs in LMICs often neglect the well-being of patients and fail to prioritize comprehensive health services. [86]. In some areas, pharmacies in LMICs often fail to provide contact details of consultants or referral materials for diseases that require specialized care. This can be a significant limitation for patients who may need further evaluation or treatment beyond what can be offered at the pharmacy level [124,128].

The overall performance of community pharmacies (CPs) in LMICs regarding medication advice and patient counseling falls below the expected standards.

Many crucial aspects of counseling are still overlooked and not given due consideration by pharmacy staff. The handling of patients, whether with or without a prescription, does not show significant differences. There are six major stages involved in patient handling: requesting medication, filling prescriptions, taking patients’histories, referring to physicians, dispensing medication, and providing advice to patients. In these stages, there are gaps and deficiencies in the level of patient care and counseling provided by CPs.

### 3.3. Determinants of Deprived Pharmacy Practices

The education system for pharmacists and pharmacy technicians across Asia is not uniform, and the content of their training programs can vary. The number of qualified personnel in the field also fluctuates from country to country. This diversity in educational backgrounds and skill sets can have a significant impact on the practices of pharmacies in different regions.

Additionally, laws, policies, and regulations governing pharmacy practices differ among countries, contributing to further variations in the way pharmacies operate. The legal framework surrounding the pharmacy profession, including licensing requirements, scope of practice, and the availability of certain medications, can greatly influence the services provided by pharmacies.

Furthermore, the presence of national-level programs and initiatives for public health can also shape the practices of pharmacies. These programs may prioritize certain areas of healthcare or focus on specific health issues, leading to variations in the services offered by pharmacies and the emphasis placed on certain aspects of patient care.

Overall, these factors, including differences in education, regulations, and public health initiatives, contribute to the diversity of pharmacy practices in Asia. It is important for regulatory bodies, policymakers, and healthcare professionals to consider these factors when working towards improving pharmacy services and ensuring optimal patient care across the region. These factors are detailed as follows.

#### 3.3.1. Knowledge of Staff

Pharmacy staff in Asian LMICs often lack knowledge about proper drug storage and dispensing practices, leading to poor performance in these areas [129,130]. The training and knowledge of drug sellers operating CPs is insufficient [131].

There is a significant knowledge gap when it comes to chronic diseases and their management among CP personnel [132]. There is a pressing need for the development of experiential-based skills and more advanced training among CP personnel. With the evolving healthcare landscape and increasing complexity of patient care, it is crucial for pharmacists and other CP staff to stay updated with the latest advancements in their field [133]. When staff members lack adequate qualifications, it can lead to poor practices and compromised patient care [134]. Clinical pharmacy education is considered to be inadequate or lacking in many settings. The training and education of pharmacists often focus more on the technical aspects of dispensing medications rather than the clinical skills required for patient care [135].The field of clinical pharmacy faces challenges in terms of the availability of well-trained staff and adequate facilities [122,136].

#### 3.3.2. Profit-Oriented Service

In CPs, revenue can be increased through three methods: maximizing client numbers, minimizing operational costs and services, and maximizing profit from each patient. However, employing these methods often leads to a decrease in the quality of services provided [86]. Pharmacies in LMICs of Asia often prioritize profit over other considerations and tend to hire low-salaried, non-qualified staff who lack the competence to promote the rational use of medicines. This profit-oriented approach hinders the promotion of appropriate and responsible medication practices [137].

To survive in the market, pharmacies in LMICs of Asia face intense competition, which puts pressure on employees to generate higher revenue [87,125,138]. On the other hand, a concerning practice is observed in India, where pharmacies hire hospital staff as their agents. These agents persuade patients who are discharged from the hospital to use the services of their affiliated CP [93]. Moreover, in order to maintain customer loyalty, the pharmacy staff often fulfills customers’ requests for any medications, irrespective of their suitability from a health and legal standpoint. Additionally, to maintain a friendly relationship with physicians, pharmacy staff refrain from questioning doctors about the appropriateness of prescribed medications. They may even dispense medications that have been prescribed inappropriately [86].

Medicine manufacturers employ aggressive marketing techniques to increase the sales of their products. They often offer attractive incentives, bonuses, and promotional offers to CPsin order to encourage them to promote and sell their products [93]. Such practices incentivize drug sellers to sell more medicines, resulting in a significant increase in turnover for both the CPsand the medicine manufacturing companies [97].

#### 3.3.3. Regulatory Intervention

Regulatory interventions are crucial in improving the practices and services of community pharmacies. They help increase the availability of essential medicines, promote patient counseling, and ensure proper dispensing practices. These interventions play a vital role in regulating and standardizing pharmacy operations for the benefit of patients and the overall healthcare system [139,140]. The regulatory infrastructure implemented by states is designed to govern and regulate the pharmaceutical sector. However, in developing countries, this infrastructure often faces significant challenges. These challenges can include limited resources, inadequate enforcement mechanisms, lack of trained personnel, and corruption. As a result, the regulatory framework may not be fully effective in ensuring compliance with standards, quality control, and patient safety. Efforts are needed to address these challenges and strengthen the regulatory infrastructure in developing countries to promote safe and effective pharmaceutical practices [141].

## 4. Conclusions

The regulation, staffing, and overall functioning of CPs require further improvementin order to enhance the quality of healthcare services. Currently, the domains of services provided by CPs are limited, and there is a need to expand and strengthen patient care services. It is crucial to ensure that medication selling is carried out in a legal and regulated manner to protect patient safety. Continuous staff training should be made compulsory to meet licensure requirements and enhance the knowledge and skills of pharmacy professionals. This will help keep them updated with the latest advancements in the field and ensure the provision of high-quality care to patients.To achieve these improvements, all Asian countries should establish a rigorous policy framework that focuses on strengthening patient care. This framework should address issues such as regulatory oversight, staff qualifications and training, scope of services, and patient safety measures. By implementing such policies, the healthcare system can be enhanced, and patients can receive comprehensive and safe care from CPs.

## 5. Limitations

This study represents the snapshot of selected countries that cannot be generalized to other countries that are not represented in this review;This paper is an opinion according to its nature;however, different methodologiesor article types may change the results and representation;Todate, the data from many countries are insufficient and cause a hindrance in the presentation of a complete picture of practices in a country.

## Figures and Tables

**Figure 1 pharmacy-11-00127-f001:**
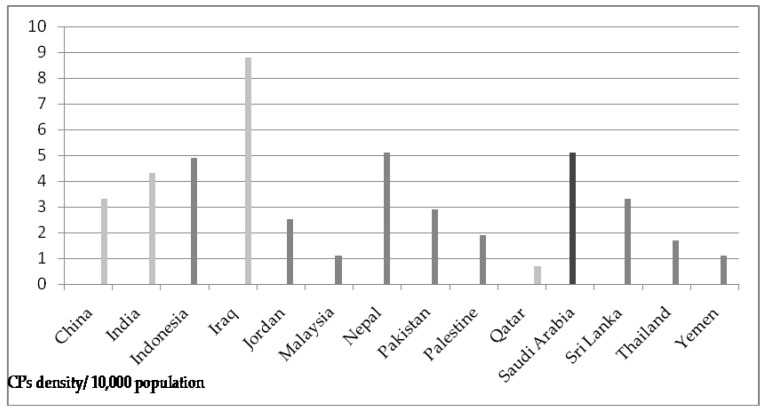
Comparison of density according to population.

**Table 1 pharmacy-11-00127-t001:** Formation and characteristics of CPs.

Sr. #	Country	Number *	Density **	Jurisdiction	PC	Traits	Challenges
1	China	472,000	3.3	Drug Administration Law, 2001	Limited	○ More common chain pharmacies ○ Sale of medicine under the supervision of a pharmacist ○ Ownership not restricted to pharmacist	○ OTC sale of antibiotics ○ Few pharmacies met the GPP certification criteria
2	India	600,000	4.3	Pharmacy Act, 1948taozhuozzzDrugs and Cosmetics Act and Rules, 1945	Very limited	○ Working of chain and individual pharmaciesis identical	○ Unqualified and non-professional ownership ○ Selling PoM without prescription
3	Indonesia	135,000	4.9	Section 51, Act 2009	Limited	○ Ownership not limited to pharmacist ○ Fivemajor chain pharmacies ○ Provision of medication and health information ○ Smoking cessation programs	○ Patients are unsatisfied
4	Iraq	9000	8.8	Pharmacy Act, 1923	Good (MTM)	○ Privately owned ○ No international chain pharmacies ○ Presence of a pharmacist is compulsory	○ Pharmacist’s performance is poorly appreciated
5	Jordan	2500	2.5	Drug and Pharmacy Law, 2001	Moderate	○ Elevenchain pharmacies ○ Chain pharmacies are better than individual pharmacy	○ Barriers toimplementation of “Medication Review” process
6	Malaysia	3300	1.1	Poison Act of 1952	Moderate	○ CPs operate as an independent, chain, or attached to clinics ○ Minimum area requirement is 200 square feet	○ Time shortage ○ Lack of human resources and manpower ○ Collaboration deficit between healthcare personnel ○ Self-inefficiency ○ Regulatory and legal constraints
7	Nepal	14,899	5.1	Drug Act, 1978	Limited	○ Very limited chain pharmacies ○ Identical working of chain and individual pharmacies ○ Pharmacist can own a single pharmacy	○ Non-pharmacists run CPs ○ Selling PoM without a prescription
8	Pakistan	63,000	2.9	Drug Act, 1976 Pharmacy Act, 1967	Very limited	○ Chain pharmacies in urban or suburban area ○ Non-pharmacists are allowed to own a pharmacy	○ Resource constraints, inadequate infrastructure, or a lack of regulatory oversight ○ Selling PoM without a prescription ○ Improper training of staff
9	Palestine	1000	1.9	Palestinian pharmacy practice law	Very limited	○ No chain pharmacy ○ Only pharmacists are allowed to own a pharmacy	○ OTC sale of antibiotics ○ Limited compounding services
10	Qatar	200	0.7	Almeezan Law No. 3, 1983	Very limited	○ Privately owned chain or individual CPs ○ CPs are under the supervision of licensed pharmacy technicians and pharmacists	○ Insufficient pharmacist ○ Limited time for patient contact ○ Low knowledge of pharmacist
11	Saudi Arabia	17,815	5.1	MOH, 1978	Limited	○ Big chains of CPs exist ○ Ownership is restricted Saudi pharmacists ○ Sale of medicine is restricted to prescription	○ Limited participation of female pharmacists
12	Sri Lanka	7029	3.3	Drugs Act No. 27, 1980	Limited	○ More chain pharmacies than individual pharmacies ○ The dispensing of medicine is allowed to registered pharmacists	○ No significant difference between the dispensing of PoM and OTC medications
13	Thailand	11,603	1.7	Thai Drug Act,1967	Limited	○ Limited number of chain pharmacies ○ Urban accumulative CPs ○ Sale of antibiotics is allowed to Class 1 medical stores ○ Chronic disease screening	○ Limited dispensing services due to dispensing rights to physicians ○ Poor enforcement and implementation of regulation
14	Yemen	3315	1.1	Law # 26 of 2002	Very limited	○ Domestic chain pharmacies	○ Sale of PoM without a prescription

* Total number of CPs, ** Density/10,000 population.

## Data Availability

Not applicable.

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
