# Peer review of "Community Pharmacies in the Asian Countries of Developing Health System: Formation, Regulation, and Implication"

_pharmacy, 2023, doi:10.3390/pharmacy11040127_

Round 1

Reviewer 1 Report

This paper represents an ambitious project to assess and compare vital characteristics of community pharmacies in 14 countries--mostly in Asia. Although the references suggest that an abundance of literature has been gathered for the project, there are significant problems in design, the uniformity of coverage for each country, lack of definition of important terms and significant errors in grammar and word choice. The subject matter is of great interest for public health and policy researchers, but the lack of consistency in the coverage of individual countries reduces the usefulness of the information for rigorous research. This reviewer is listing some specific comments but many more could be made.

Authors. Author information is incomplete. Countries of origin are not noted.

Abstract. Lacks information about purpose and methodology.

Throughout the text, terms are used without definition; e.g., narrative review, grey literature, traditional services, low/middle income country, "below the bar", medication management therapy, etc.

Errors in terms of word choice particularly in Section 3.2: (line 201)--"rottenly" instead of "routinely", line 206--"deprived" performances, etc.

Many instances of disagreement of subject and verb relative to singular/plural.

In the Methods, failure to describe the criteria for inclusion and exclusion, what data are extracted from each study, etc. In the descriptions of individual country's CPs, coverage varies in depth and issues addressed. There could be an issue of bias against LMICs countries although this would be hard to prove. This reviewer understands the difficulty of making valid comparisons with the limited information available, but that difficulty must be acknowledged not ignored.

References are rich as a resource but the formatting of individual references lacks consistency. One reference checked did not pull up a study.

Overall, the paper has merit in providing a snapshot of community pharmacy systems in the 14 countries studied. If possible, the authors need to go back to the information sources and define the parameters they are using to characterize each country's CP system. In terms of identifying "flaws" (Section 3.2), there is limited usefulness in "lumping" the information from multiple sources together. It would be more useful to identify countries doing a good job and/or those with significant problems. 

Comments are included in the section above.

Author Response

This paper represents an ambitious project to assess and compare vital characteristics of community pharmacies in 14 countries--mostly in Asia. Although the references suggest that an abundance of literature has been gathered for the project, there are significant problems in design, the uniformity of coverage for each country, lack of definition of important terms and significant errors in grammar and word choice. The subject matter is of great interest for public health and policy researchers, but the lack of consistency in the coverage of individual countries reduces the usefulness of the information for rigorous research. This reviewer is listing some specific comments but many more could be made.

Authors. Author information is incomplete. Countries of origin are not noted.

Thank you so much for your professional and valuable comments  to improve the quality of manuscript. Each  proportion is addressed as advised

Abstract. Lacks information about purpose and methodology.

Throughout the text, terms are used without definition; e.g., narrative review, grey literature, traditional services, low/middle income country, "below the bar", medication management therapy, etc.

Ans : Methodology is added

Terms are also included

Errors in terms of word choice particularly in Section 3.2: (line 201)--"rottenly" instead of "routinely", line 206--"deprived" performances, etc.

Many instances of disagreement of subject and verb relative to singular/plural.

Ans :   corrected as advised

In the Methods, failure to describe the criteria for inclusion and exclusion, what data are extracted from each study, etc. In the descriptions of individual country's CPs, coverage varies in depth and issues addressed. There could be an issue of bias against LMICs countries although this would be hard to prove. This reviewer understands the difficulty of making valid comparisons with the limited information available, but that difficulty must be acknowledged not ignored.

Ans :  Inclusion and exclusion criteria are added.

A section of study limitations is added for further clarification

References are rich as a resource but the formatting of individual references lacks consistency. One reference checked did not pull up a study.

Ans :   corrected as advised

Overall, the paper has merit in providing a snapshot of community pharmacy systems in the 14 countries studied. If possible, the authors need to go back to the information sources and define the parameters they are using to characterize each country's CP system. In terms of identifying "flaws" (Section 3.2), there is limited usefulness in "lumping" the information from multiple sources together. It would be more useful to identify countries doing a good job and/or those with significant problems

Ans :   Terms are modified and corrected

Once again very grateful to you for your valuable time and suggestions. Regards

Reviewer 2 Report

Dear Authors,

Thank you for the opportunity to read and review the article titled “Community pharmacies in the Asian countries of developing health system: Formation, regulation and implication” for the journal Pharmacy. This is an opinion paper with a narrative review method, that presents descriptive characteristics of community pharmacies regulation, organization, and practices in Asian countries. I find the article interesting and informative, and should be of interest to the journals’ readers. However, I suggest some minor revisions to the manuscript before publication.

I hope my comments are helpful!

Abstract:

I recommend adding the aim of the paper and a short description of the method.

Introduction:

I would recommend to add an aim of this opinion paper; why did you choose to perform this narrative review and write this paper?

Method:

Line 58-60 could be moved to the end of the introduction.

Where and how did you search for grey literature?

Did you limit the literature search to any time-frame? Since you are describing features of pharmacy services and practices, it is relevant to know the years the literature represents. What were the eligibility criteria for the studies (besides being written in English?)

What kind of data were you planning to extract from the literature? Please provide the reader with some information on data items that were sought.

Results:

What is the evidence base for this review? How many papers did you include from the searches, and what knowledge sources were included from grey literature?

In general, I was curious of why and how the countries were selected for review, e.g., China, India, Indonesia, and so on. What about the others? I would also recommend some kind of consistent reporting of results, and that point is also linked to my question on which data you planned to retrieve/extract in your review synthesis. For example, for Palestine, you provide information about most qualified pharmacist being female. Then I got curious, how is gender distribution amongst pharmacists in the other countries? For the various countries presented, there are not equivalent information presented, making it hard to compare, if that is desirable.

Line 155: write KSA as Kingdom of Saudi Arabia (KSA), or introduce the abbreviation and correct name in the sub-headline (line 154).

Line 235: Different classes of drug stores are described in Thailand. This information should be included earlier (in subchapter 3.1.13. Thailand)?

Line 266-268: This sentence is unclear to me, could you try to rephrase to convey the meaning in a clearer way? Include a reference?

In the results, the number of pharmacies per country is presented. I would recommend adding information about pharmacy density, e.g., the number of pharmacies per 10 000 citizens, making it easier for the reader to compare the pharmacy numbers between countries.

Due to the topic of this paper, it would be of interest to read about which authority regulates the community pharmacy industry in these Asian countries, since there appears to be deviations that needs attention. This is nicely presented for e.g., Indonesia and India, but not for all countries mention in the review. Regulatory interventions are suggested in conclusion, so it would be feasible to describe status quo.

Another topic that I really would like to read about in such a paper, is the status of the online pharmacy industry. How is this organized and regulated, and what are the predictions for this segment of the pharmacy industry for the future in Asia?

Conclusions

I would recommend to emphasize that this conclusion is relevant for community pharmacies in Asian countries.

References

The references are not numbered in the reference list, so it is impossible to review. The reference list must be updated with numbers according to Pharmacy reference style.

I would recommend English editing of the entire manuscript, and revise the use of terms. For example the use of the terms “drugs”, “medicines” and “medications” interchangeably. I would recommend consequent use of one term throughout the manuscript (unless the terms have different meaning to you).

Clean the document for redundant punctum and spaces.

Author Response

Thank you for the opportunity to read and review the article titled “Community pharmacies in the Asian countries of developing health system: Formation, regulation and implication” for the journal Pharmacy. This is an opinion paper with a narrative review method, that presents descriptive characteristics of community pharmacies regulation, organization, and practices in Asian countries. I find the article interesting and informative, and should be of interest to the journals’ readers. However, I suggest some minor revisions to the manuscript before publication.

I hope my comments are helpful!

Thank you so much for your professional and valuable comments  to improve the quality of manuscript. Each  proportion is addressed as advised

Abstract:

I recommend adding the aim of the paper and a short description of the method.

Ans : Methodology is added

Introduction:

I would recommend to add an aim of this opinion paper; why did you choose to perform this narrative review and write this paper?

Ans :  Study rationale is added

Method:

Line 58-60 could be moved to the end of the introduction.

Where and how did you search for grey literature?

Did you limit the literature search to any time-frame? Since you are describing features of pharmacy services and practices, it is relevant to know the years the literature represents. What were the eligibility criteria for the studies (besides being written in English?)

What kind of data were you planning to extract from the literature? Please provide the reader with some information on data items that were sought.

Ans :  Inclusion and exclusion criteria are added.

Results:

What is the evidence base for this review? How many papers did you include from the searches, and what knowledge sources were included from grey literature?

In general, I was curious of why and how the countries were selected for review, e.g., China, India, Indonesia, and so on. What about the others? I would also recommend some kind of consistent reporting of results, and that point is also linked to my question on which data you planned to retrieve/extract in your review synthesis. For example, for Palestine, you provide information about most qualified pharmacist being female. Then I got curious, how is gender distribution amongst pharmacists in the other countries? For the various countries presented, there are not equivalent information presented, making it hard to compare, if that is desirable.

Ans :   as it a view point and opinion type of article , no criteria for retrieve of data was applied.

Line 155: write KSA as Kingdom of Saudi Arabia (KSA), or introduce the abbreviation and correct name in the sub-headline (line 154).

Ans :   corrected as advised

Line 235: Different classes of drug stores are described in Thailand. This information should be included earlier (in subchapter 3.1.13. Thailand)?

Ans :   corrected as advised

Line 266-268: This sentence is unclear to me, could you try to rephrase to convey the meaning in a clearer way? Include a reference?

Ans :   corrected as advised

In the results, the number of pharmacies per country is presented. I would recommend adding information about pharmacy density, e.g., the number of pharmacies per 10 000 citizens, making it easier for the reader to compare the pharmacy numbers between countries.

Ans :   corrected as advised

Conclusions

I would recommend to emphasize that this conclusion is relevant for community pharmacies in Asian countries.

Ans :   corrected as advised

References

The references are not numbered in the reference list, so it is impossible to review. The reference list must be updated with numbers according to Pharmacy reference style.

Ans :   corrected as advised

Once again very grateful to you for your valuable time and suggestions. Regards

Round 2

Reviewer 1 Report

English language issues are much improved in the Abstract and Introduction.

Authors are still not identified by country. Purpose not stated in the Abstract. Many terms are still not defined; e.g., grey literature, Pharmaceutical Care, etc. 

Methods. Missing are dates about time frame, who did the search, how disagreements about inclusion, exclusion, etc were addressed. 

Challenges section still has many grammatical errors and problems with work choice (...various drugs are muddle up...) and sentence structure. Same comments for Section 3.2.5 through 3.3.3.

Conclusions. Needs work on being specific and clear--what does medication selling should be legal mean?

References are still not consistently formatted.

One or two sections have been revised. The majority of the paper doesn't appear changes and still requires work.

Author Response

English language issues are much improved in the Abstract and Introduction.

Very grateful to you for your valuable time. The grammatical mistakes are corrected in the other portion . Please see the current version.

Authors are still not identified by country.

We can understand this question. Is it meant that we make a table of studies as in the systematic review ?

If yes. Our study design adopts narrative methodology.

 Purpose not stated in the Abstract.

Included in the current version. as given below

"The aim of this study was to conduct a comprehensive literature review in order to enhance understanding of the establishment and regulation of CPs. By undertaking this review, the researchers, policy makers and practitioners sought to gain a deeper insight into the performance and functions of CPs and the regulatory frameworks that govern them. Moreover; this review shad a light on implementation strategies, effects on patient outcomes, and the barriers and challenges associated with their establishment"

Many terms are still not defined; e.g., grey literature, Pharmaceutical Care, etc.

3 terms are used in the paper

  1. Good Pharmacy Practice

Please see the line # 2 section 3.1.8.  

It involves ensuring the proper handling, storage, dispensing, and counseling of medications, as well as maintaining high standards of professionalism and patient care

  1. grey literature

Please see the line # 3 of  section 2

grey literature (government documents. evaluations, reports, white papers and working papers)

  1. Pharmaceutical Care

Please see the line # 13 of  section 3.1.8.  

PC encompass the patient-centered MTM for optimal outcomes

Methods. Missing are dates about time frame, who did the search, how disagreements about inclusion, exclusion, etc were addressed. 

Included in the current version. Please see the section 2, as given below

  1. The records were extracted from Nov 2022 to May 2023

Included in the current version Please see the section 2. as given below

The studies identified through the literature search were screened by three groups independent reviewers to assess their eligibility. Group A (SA, MA) was responsible for record gathering for each country. Group B (RA, GA) was responsible to identify the literature about current performance. Group C (ROA, EA) was responsible for study screening about determinants Any disagreements were resolved through consensus or by a third member from group of reviewer (MQ, WS).

Challenges section still has many grammatical errors and problems with work choice (...various drugs are muddle up...) and sentence structure. Same comments for Section 3.2.5 through 3.3.3.

The grammatical mistakes are corrected in the other portion . Please see the current version

Conclusions. Needs work on being specific and clear--what does medication selling should be legal mean?

Its rewritten. Please see the current version, the section 2. as given below

The regulation, staffing, and overall functioning of CPs require further improvement in order to enhance the quality of healthcare services. Currently, the domains of services provided by CPs are limited, and there is a need to expand and strengthen patient care services. It is crucial to ensure that medication selling is carried out in a legal and regulated manner to protect patient safety. Continuous staff training should be made compulsory to meet licensure requirements and enhance the knowledge and skills of pharmacy professionals. This will help keep them updated with the latest advancements in the field and ensure the provision of high-quality care to patients. To achieve these improvements, all Asian countries should establish a rigorous policy framework that focuses on strengthening patient care. This framework should address issues such as regulatory oversight, staff qualifications and training, scope of services, and patient safety measures. By implementing such policies, the healthcare system can be enhanced, and patients can receive comprehensive and safe care from CPs.

References are still not consistently formatted.

Corrected as advised .

Thanks and Best Regards